# The long non-coding RNA *Dreg1* is required for optimal ILC2 development

**Sara Quon[1,2†], Adelynn Tang[3,4,5†], Nadia Iannarella[1,2], Kael Schoffer[1,2], Wing Fuk Chan[1,2], Timothy M Johanson[2], Ajithkumar Vasanthakumar[3,4,5*‡], Rhys Allan[1,2*‡]**

[1]The Walter and Eliza Hall Institute of Medical Research, Parkville, Australia; [2]Department of Medical Biology, The University of Melbourne, Parkville, Australia; [3]Olivia Newton-John Cancer Research Institute, Heidelberg, Australia; [4]La Trobe University, Bundoora, Australia; [5]Peter MacCallum Cancer Centre, Melbourne, Australia

**\*For correspondence:**
ajith.vasanthakumar@petermac.org (AV);
rallan@wehi.edu.au (RA)

[†]These authors contributed equally to this work
[‡]These authors also contributed equally to this work

**Competing interest:** The authors declare that no competing interests exist.

## eLife Assessment

This is a **valuable** study that investigates the role of the long non-coding RNA Dreg1 for the development, differentiation, or maintenance of group 2 ILC (ILC2). The authors generate Dreg1-/- mice and show **solid** evidence for a reduction of group 2 innate lymphoid cells (ILC2). However, the strength of evidence supporting and analysing the impact of Dreg1 on Gata3 expression, a transcription factor required for ILC2 cell fate decisions, remains incomplete. This study will be of interest to immunologists.

**Abstract** Gata3 is an essential transcription factor for the development of several distinct immune cell lineages such as T cells, natural killer (NK) cells, and innate lymphoid cells (ILCs). As such, the levels and timing of *Gata3* expression are critical for directing lineage fate decisions. The *Gata3* locus has a complex and dynamic distal regulatory enhancer landscape. Recently, we identified a non-coding RNA, *Dreg1*, located immediately upstream of the classic +280 kb T/NK cell enhancer (Tce1). To test its function, we excised the *Dreg1* locus in mice and observed a selective reduction of group 2 ILCs (ILC2) across multiple tissues, but mature T, NK, and other ILC lineages remained unchanged. In bone marrow, common innate lymphoid cell progenitors (ILCPs) increased while ILC2 progenitors (ILC2P) decreased, with a modest reduction of *Gata3* in upstream progenitors consistent with an early developmental bottleneck. Chromatin profiling showed the Dreg1 locus is accessible in early lymphoid progenitors and became decorated with H3K27ac in ILCP in a Tcf1-dependent manner. Furthermore, Tcf1-deficient cells did not express *Dreg1* and showed alterations in the epigenetic landscape of the *Dreg1* locus. Finally, we discovered that potential homologues of *Dreg1* harboured in a syntenic enhancer of *GATA3* are also highly expressed in human ILC2. Taken together, we conclude that *Dreg1* is a Tcf1-dependent non-coding RNA critical for fine tuning the high level of *Gata3* required for the optimal development of the ILC2 lineage.

## Introduction

The immune system comprises a collection of functionally distinct cell types which act in concert to protect the body from infection and malignancy. These diverse cell populations develop from common haematopoietic progenitors in a highly controlled fashion driven by a network of transcription factors that control lineage-specific gene expression programs. GATA-binding protein 3 (Gata3) is one of the master transcription factors that regulates the development of immune cells such as T

cells, Natural Killer (NK) cells, and innate lymphoid cells (ILCs) (*Harly et al., 2018*; *Ho et al., 2009*). Gata3 is also critical for the development of CD4[+] T helper 2 (Th2) cells (*Zheng and Flavell, 1997*) which produce cytokines such as IL4, 5, and 13 that are critical for protection against extracellular pathogens. It has recently emerged that a population of ILCs known as group 2 ILCs (ILC2) with similar characteristics to Th2 cells reside in non-lymphoid tissues such as the small intestine, visceral adipose tissue and lung and have been shown to play a non-redundant role in immunity to helminths and drive allergic immune responses (*Jarick et al., 2022*). ILC2s develop in the bone marrow via common ILC progenitor (CILP) (*Yu et al., 2014*), common helper ILC progenitor (CHILP) (*Klose et al., 2014*), innate lymphoid cell progenitor (ILCP) (*Constantinides et al., 2014*), and, finally, ILC2 progenitor (ILC2P) (*Hoyler et al., 2012*). Gata3 is critical to the development of ILCs (*Hoyler et al., 2012*; *Yagi et al., 2014*), and high levels are required for the development of ILC2P in the bone marrow (*Hoyler et al., 2012*; *Klein Wolterink et al., 2013*).

Dosage of Gata3 plays a critical role in T, NK, and ILC development and peripheral immunity (*Hoyler et al., 2012*; *Klein Wolterink et al., 2013*; *Scripture-Adams et al., 2014*). To support this, *Gata3* expression is spatiotemporally tuned by several distal enhancers that dictate *Gata3* activity in these individual lineages. Previous research identified a 7.1 kb region ~280 kb from the *Gata3* gene that is important for its expression in NK and T cell development, known as T cell enhancer 1 (Tce1) (*Hosoya-Ohmura et al., 2011*; *Ohmura et al., 2016*). Subsequent work found that this region (also termed *Gata3* +278/285) also impacts ILC2 development but not Th2 cell differentiation (*Kasal et al., 2021*). ILC2-specific effects have been attributed to additional distal regulatory regions, such as *Gata3* +674/762 (*Kasal et al., 2021*) and the Gata3 tandem super enhancers (+500/764) (*Furuya et al., 2024*). In addition, a Th2-specific enhancer (*Gata3* +906/935) has recently been identified (*Kumagai et al., 2024*). Overall, this complex regulatory landscape mechanistically fits with a dosage-threshold model controlling *Gata3* expression.

Recently, we discovered a long non-coding RNA gene that we named *Distal regulatory enhancer of Gata3 1 (Dreg1)* which lies ~0.6 kb upstream of Tce1 enhancer (*Chan et al., 2021*; *Chan et al., 2022*). Expression of *Dreg1* is high in early T cell progenitors and Th2 cells, mirroring *Gata3* expression (*Chan et al., 2021*). Overexpression studies indicated that it may be involved in the establishment but not maintenance of *Gata3* expression (*Chan et al., 2021*; *Chan et al., 2022*), but it remains unclear how the loss of *Dreg1* affects immune system development. Here, we explore the role of *Dreg1* in the regulation of immune cell differentiation. We find that deletion of the *Dreg1* locus leads to a specific loss of ILC2 progenitors and their downstream progeny in non-lymphoid tissues but not T or NK cells. Chromatin profiling reveals that the regulatory regions around the *Dreg1* gene are open in haematopoietic progenitors and are bound by the transcription factor TCF1 in early innate lymphoid progenitors and TCF1-deficient mice lack *Dreg1* expression. Finally, we discovered potential homologues of *Dreg1* that are highly expressed in human ILC2 emanating from a syntenic enhancer of GATA3.

## Results

### Deletion of the *Dreg1* locus results in a specific reduction of ILC2

To understand how *Dreg1* contributes to the development of the immune system, we generated *Dreg1*-deficient mice by CRISPR-Cas9-mediated excision of the complete 3 kb of the *Dreg1* gene (*Figure 1A*). We confirmed that the *Dreg1* gene was removed from the germline and found that these mice reproduced at Mendelian frequencies and exhibited no overt developmental defects suggesting that *Dreg1* is not required for embryogenesis.

Given our previous finding of the relatively high levels of *Dreg1* expression in T cells (*Chan et al., 2021*), and of a critical role of the Tce1 (*Gata3* +278/285) enhancer (*Hosoya-Ohmura et al., 2011*; *Kasal et al., 2021*), we expected that the deletion of *Dreg1* would impact T cell and NK cell development. We therefore performed an analysis of the different populations of T cells. Mature CD4[+] and CD8[+] T cells in the spleen appeared unaltered in frequency (*Figure 1—figure supplement 1*). Moreover, both populations had similar proportions of naïve, effector, and memory cells (*Figure 1—figure supplement 1B and C*). This suggests that *Dreg1* is not required for conventional T cell development or maintenance. In addition, the loss of *Dreg1* did not affect the development of NK cells and γδ T cells as normal proportions were observed in the spleen (*Figure 1—figure supplement 1D*). Overall, this analysis suggests that *Dreg1* is dispensable for NK and T cell development.

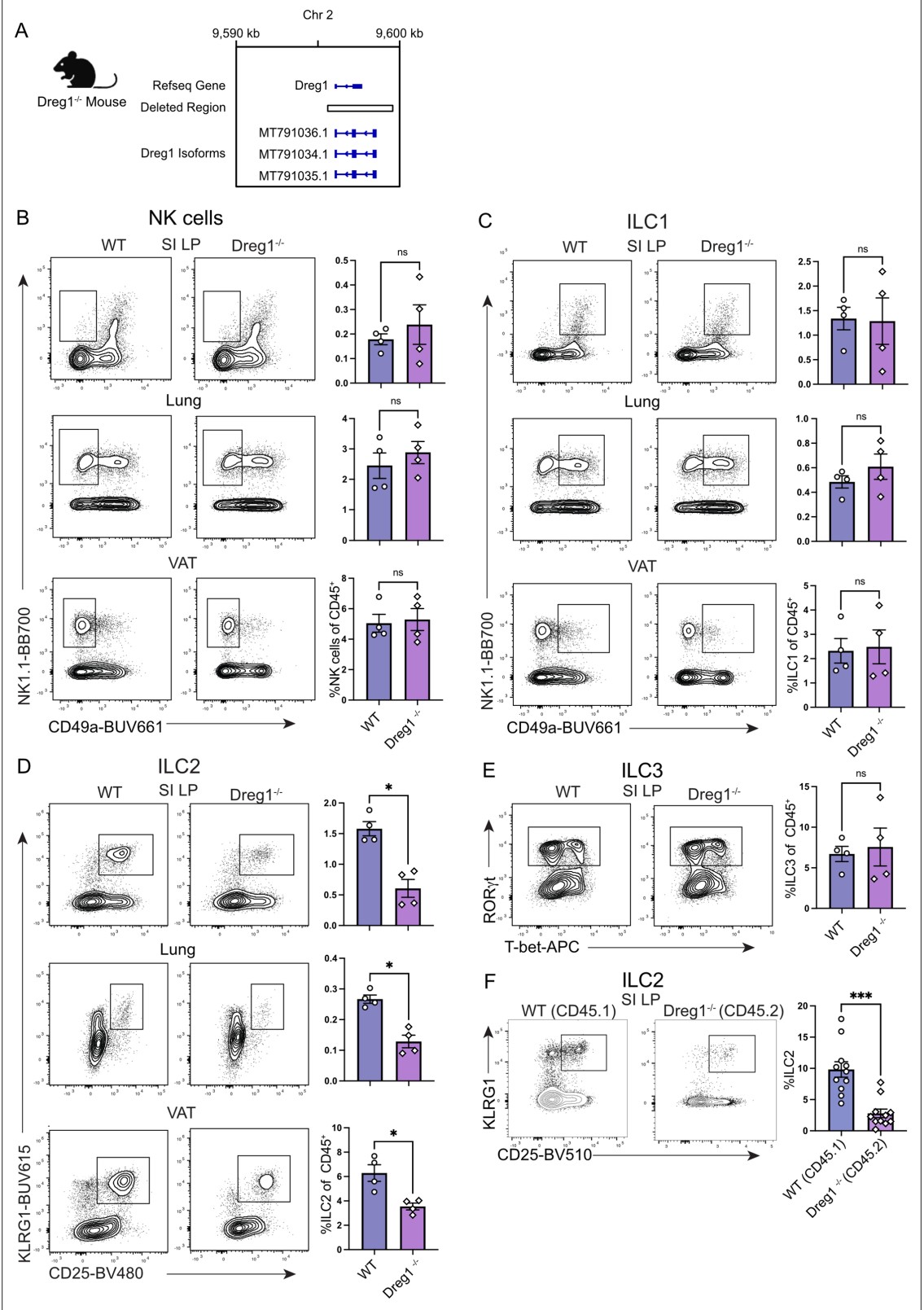

**Figure 1.** *Dreg1* deletion results in a specific reduction in peripheral ILC2 cells. (**A**) Deletion of Dreg1 in mice using CRISPR-Cas9. (**B–E**) Representative FACS plots and % of CD45+, TCRβ-, CD19-, CD11c- NK, and ILC populations from the visceral adipose tissue (VAT), small intestinal lamina propria (SI LP), and lung from wild type (WT) or Dreg1-/- mice. Shown is one representative experiment of two with n=4 mice/group. (**F**) SI LP from mixed bone marrow

*Figure 1 continued on next page*

*Figure 1 continued*

chimeras were examined for the proportion of ILC2 from the wildtype (CD45.1) or Dreg1-deficient (CD45.2) compartment. Data is pooled from three independent experiments. Mean and SEM together with individual data points are shown. Data were statistically analysed by Student's *t*-test.

The online version of this article includes the following figure supplement(s) for figure 1:

**Figure supplement 1.** *Dreg1* deletion does not affect T or NK cells.

Gata3 is also critical for the development of all innate lymphoid cells (ILCs) (*Yagi et al., 2014*), and the levels of Gata3 are critical in directing ILC2 lineage development (*Hoyler et al., 2012*; *Zhong et al., 2020*). Therefore, we next explored peripheral sites known to harbour substantial numbers of these cell types such as visceral adipose tissue (VAT), the small intestine lamina propria (SI LP), and the lung. While these tissues had relatively normal proportions of NK, ILC1, and ILC3 (*Figure 1B, C and E*), we noted a specific reduction in the population of ILC2 in each of the tissues we examined (*Figure 1D*). Thus far, the data obtained is from mice in which Dreg1 is deleted in all cells, hampering the ability to determine whether the loss of Dreg1 intrinsically regulates ILC2 development or the phenotype is driven by extrinsic signals. To demonstrate a cell-intrinsic role of Dreg1 in ILC2s, we generated mixed bone marrow irradiation chimeras comprising a ratio of 50:50 of wild type (CD45.1[+]) to Dreg1-deficient (CD45.2[+]) cells. In these mice, we observed a reduction in Dreg1-deficient ILC2 in comparison to their wild type ILC2 counterparts in all tissues examined (*Figures 1E–G*). In contrast, we observed comparable proportions of wild type and Dreg1-deficient B cells from these same tissues (*Figure 1—figure supplement 1F and G*). Taken together, these results show that the loss of *Dreg1* results in the specific and intrinsic loss of ILC2 in different peripheral tissues.

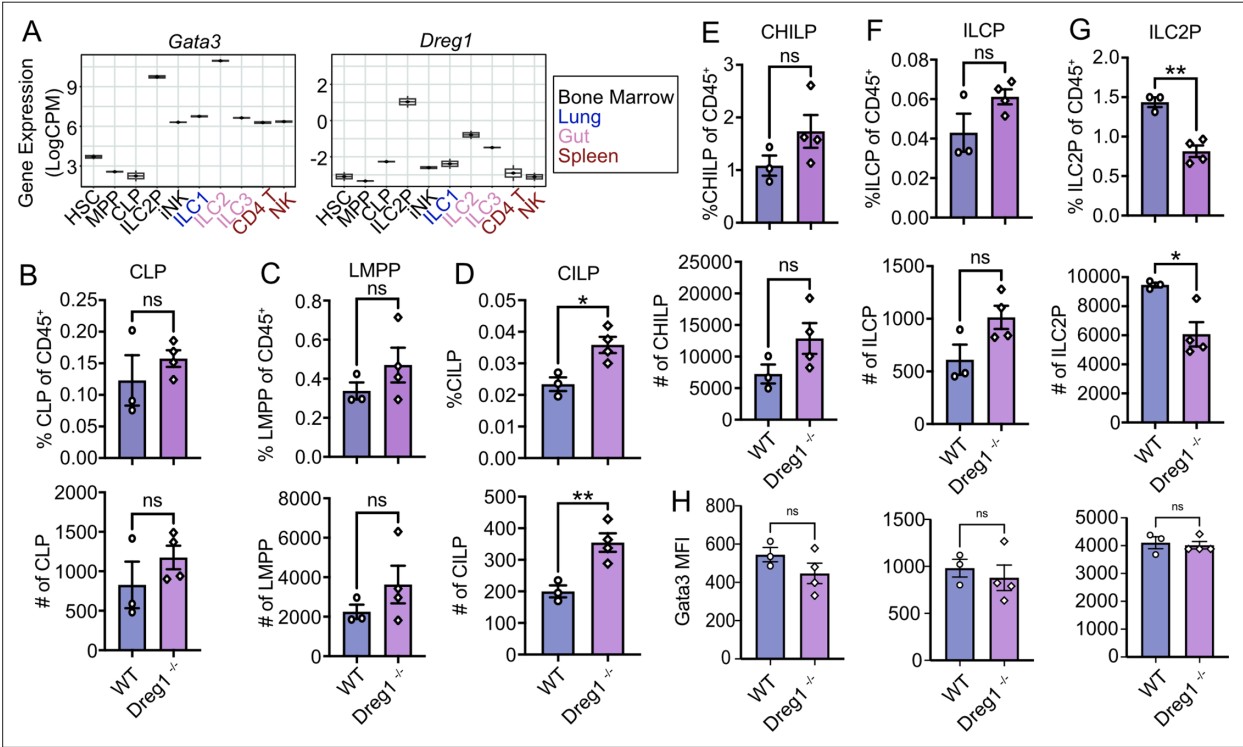

**Figure 2.** *Dreg1* deletion results in a bottleneck in ILC development and a reduction in ILC2P in the bone marrow. (**A**) Boxplots showing expression of *Gata3* and *Dreg1* from RNA-Seq (GSE77695). (**B–G**) Quantification of ILC progenitor subsets in the bone marrow with percentages (upper) and total numbers (below). (**H**) Mean fluorescence intensity (MFI) of Gata3 measured on CHILP, ILCP, and ILC2P. Shown is one representative experiment of two with n=3–4 mice/group. Mean and SEM together with individual data points are shown. Data were statistically analysed by Student's *t*-test.

The online version of this article includes the following figure supplement(s) for figure 2:

**Figure supplement 1.** Gating of ILC progenitors in bone marrow.

## *Dreg1* deficiency results in the reduction of ILC2 progenitors

We next investigated the origins of the ILC2 defect in the *Dreg1*-deficient mice. It is known that expression of *Gata3* in the early ILC progenitor stage is required for development of mature ILC2s (*Hoyler et al., 2012*; *Klein Wolterink et al., 2013*; *Zhong et al., 2020*). Indeed, analysis of publicly available RNAseq data *Shih et al., 2016* demonstrated that Gata3 expression was very high in ILC2P and ILC2 compared to other populations, whereas the highest expression of Dreg1 was found in ILC2P (*Figure 2A*). We examined ILC development in the bone marrow (*Figure 2B–G*, gating strategy in *Figure 2—figure supplement 1A*). We began by examining the common lymphoid (CLP) and lymph-myeloid primed progenitors (LMPP) and found no overt changes in proportions or total number of these populations (*Figure 2B and C*). However, we found increased CILP, CHILP and ILCP in the *Dreg1*-deficient mice (reaching significance in the CILPs) (*Figure 2D–F*). We observed a substantial reduction of the ILC2P population suggesting that the loss of *Dreg1* leads to a block in ILC development (*Figure 2G*). This suggests that the loss of *Dreg1* results in a bottleneck at the early stages of ILC development. We next examined the levels of Gata3 in these populations (*Figure 2H*). We observed

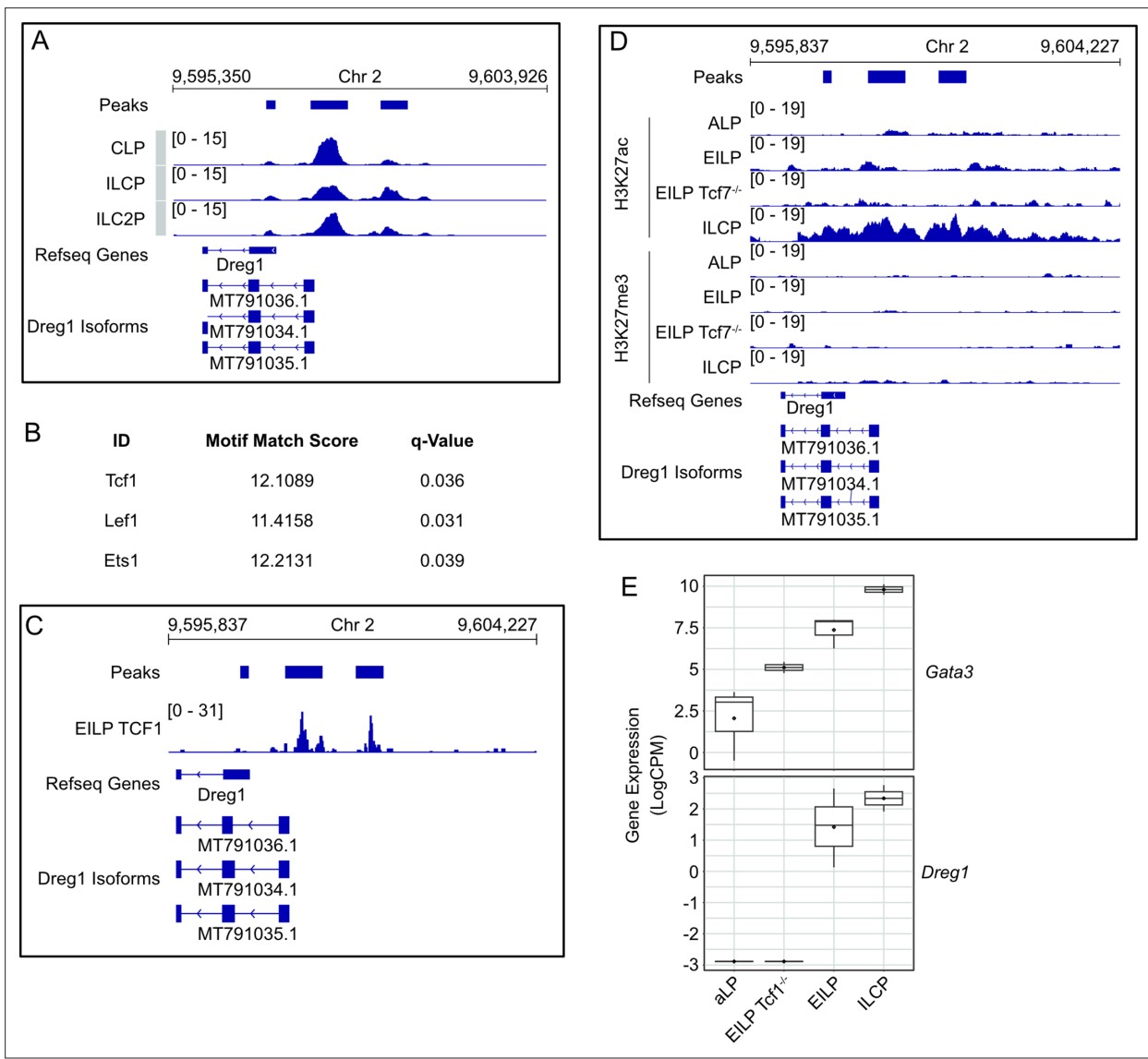

**Figure 3.** *Dreg1* locus is dynamically regulated in a Tcf1-dependent manner during ILC2 development. (**A**) Chromatin accessibility around Dreg1 in ILC progenitors (GSE169542). (**B**) Motifs enriched in accessible regions around Dreg1. (**C**) Tcf1 binding around Dreg1 locus in EILP (GSE128483). (**D**) Histone marks around Dreg1 at different stages of ILC development and in Tcf7-/- EILP (GSE142468). (**E**) *Gata3* and *Dreg1* expression quantified from RNA-Seq data in ILC progenitors and Tcf7-/- EILP (GSE113767).

a reduction in Gata3 signal in the upstream CHILP and the ILCP from the *Dreg1*-deficient mice, but interestingly Gata3 levels were normal in the ILC2P, even though this population is reduced in the *Dreg1*-deficient mice. This suggests that *Dreg1* acts to regulate Gata3 prior to the ILC2P stage which leads to a block in their development; however, if cells can overcome this perturbation, they express relatively normal levels of Gata3.

## Dynamic epigenetic regulation of the *Dreg1* locus in a Tcf1-dependent manner during ILC development

To gain a greater understanding of how the expression of *Dreg1* is regulated, we next studied the chromatin state of the *Dreg1* locus during early ILC development via analysis of publicly available ATAC-Seq data generated on progenitors and their downstream ILC progeny (*Kasal et al., 2021*; *Figure 3A*). Of note, the original Refseq annotation of Dreg1 was missing the first exon and TSS which we identified (*Chan et al., 2021*) and show in *Figure 3A*. We observed three significant accessible chromatin 'peaks', with the predominant one at the promoter which was open at the CLP stage remained so in ILCP and ILC2P. One region downstream became more accessible in ILCP and ILC2P in line with that observed by others (*Kasal et al., 2021*). Overall, we find the promoter of *Dreg1* is open in early lymphoid progenitors and the acquisition of chromatin accessibility downstream correlates with increased *Dreg1* expression in ILC2 progenitors.

To understand the transcription factors that might be involved in regulating *Dreg1* expression, we examined the motifs at the *Dreg1* promoter and downstream accessible region. We observed a specific enrichment of motifs for Tcf1, Lef1 and Ets1, suggesting that these transcription factors may be involved in regulating *Dreg1* expression. As Tcf1 (encoded by the *Tcf7* gene) has been implicated in ILC2 development (*Mielke et al., 2013*; *Yang et al., 2015*), we focussed on its role in regulating *Dreg1* expression. Firstly, we examined publicly available Tcf1 CUT&Run data (*Harly et al., 2019*) and found that indeed Tcf1 was specifically bound to the accessible sites of the *Dreg1* locus in early innate lymphoid progenitors (EILP, which are a population that sit between CILP and CHILP) (*Figure 3B*). We next examined ChIC-Seq data (*Ren et al., 2022*) of active (H3K27ac) or repressive (H3K27me3) histone modifications at different stages of ILC development (*Figure 3D*). Firstly, we found no evidence of H3K27me3 in this region in any of the cell types examined. However, there was a modest accumulation of H3K27ac around the accessible regions in the *Dreg1* locus at the EILPs which was noticeably increased at the ILCP stage. Of note, Tcf1-deficient (via deletion of the *Tcf7* gene) EILP showed lower levels of H3K27ac, suggesting that Tcf1 is likely involved in creating a permissive chromatin landscape at the *Dreg1* locus during ILC development. Indeed, the expression of *Dreg1* was dependent on the presence of Tcf1 as *Tcf7*-/- EILPs lacked *Dreg1* expression (*Harly et al., 2019*; *Figure 3E*). Taken together, this data suggests that the alterations to the chromatin state *Dreg1* locus and its subsequent transcriptional activity during ILC development is dependent on Tcf1.

## Identification of a human GATA3 enhancer harbouring *Dreg1* homologues that are active in ILC2s

Previously, we identified a region syntenic to the mouse *Dreg1* locus on human chromosome 10 that is a similar distance from the *GATA3* gene and forms a 3D interaction in T cells but not B cells, suggesting it may represent an enhancer element (*Chan et al., 2021*). Interestingly, bidirectional transcription initiates from the sequence-conserved *cis* element to produce two potential human homologs of *Dreg1*, the lncRNAs CAT00000105356.1 (we named *DREG1.1*) and CAT00000117261.1 (*DREG1.2*). To explore whether these transcripts are also expressed in ILC populations, we examined publicly available RNAseq data from healthy human blood (*Ercolano et al., 2020*; *Figure 4A*). We found that GATA3 was highly expressed in human ILCP, ILC2, and TH2 cells. We also observed this pattern for *DREG1.1* and to an even greater extent for *DREG1.2,* suggesting that the syntenic region harbours transcripts which reflect the same expression pattern as murine *Dreg1*.

Finally, we investigated the functional role of this potential enhancer of GATA3 by examining a recently published tiling CRISPR deletion screen searching for regulatory elements that control GATA3 expression in human Th2 cells (*Chen et al., 2023*). We focused on the syntenic region which was defined as functional sequence (FS) 23 (*Figure 4B*). Specifically, guides that deleted conserved sequences FS23-4 and FS23-5 led to a significant downregulation of GATA3 expression (*Chen et al., 2023*), suggesting that this region represents a distal enhancer of GATA3 (*Figure 4B*). ATACseq data

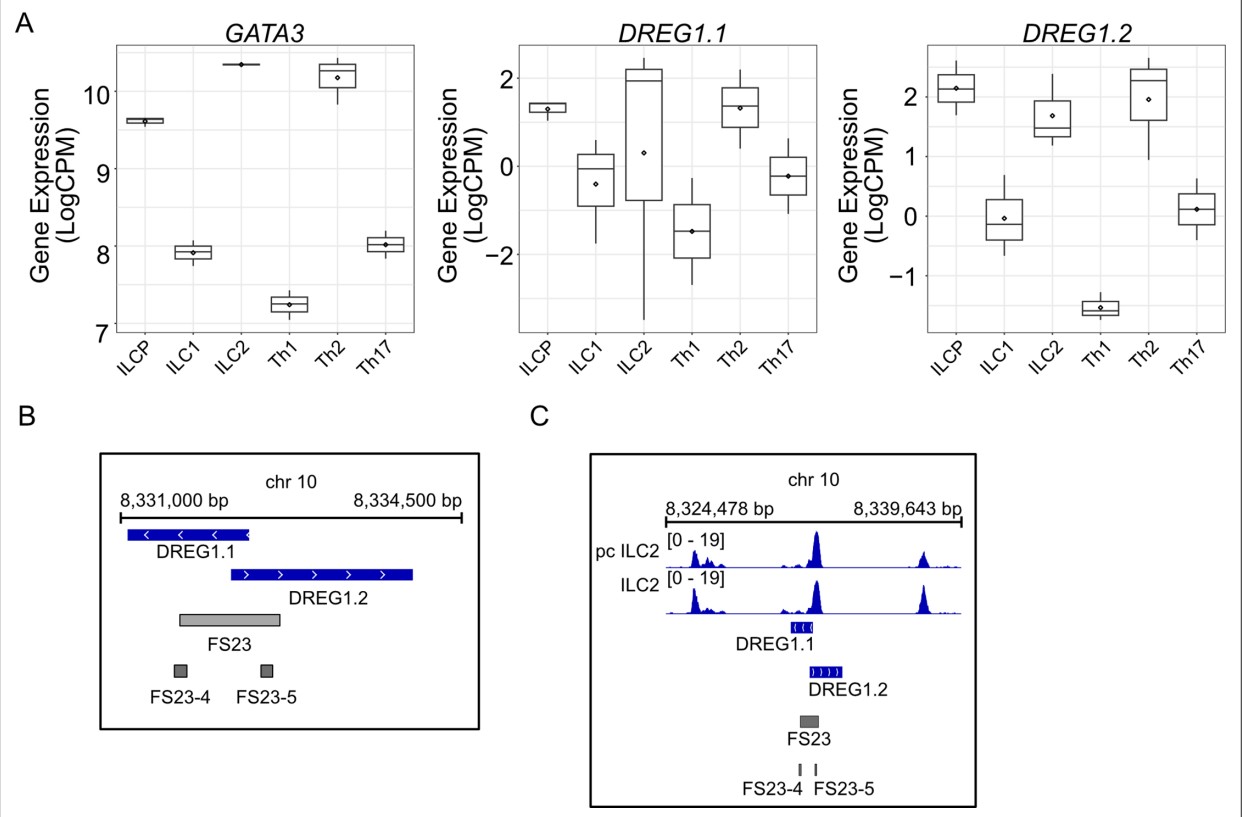

**Figure 4.** A syntenic region in humans represents a *GATA3* enhancer element with a transcriptional profile akin to murine *Dreg1*. (**A**) *GATA3* and *DREG1.1 and DREG1.2* expression quantified from RNA-Seq data in ILC and T helper populations isolated from healthy human blood (PRJEB35186). (**B**) Alignment of Functional Sequence 23 (FS23) that overlaps the syntenic region revealed as regulating GATA3 expression in a CRISPR deletion screen of human TH2 cells (***Chen et al., 2023***). (**C**) Chromatin accessibility around FS23 guides from the CRISPR deletion screen (***Chen et al., 2023***) (GSE231999).

from either cultured (pcILC2) or fresh ILC2 from healthy human blood (***Laurie et al., 2024***) revealed a clear accessible region in these cells that overlaps the *DREG1.1/2* and FS23 (***Figure 4C***). Overall, we find that this syntenic region in humans represents an enhancer that is important for high levels of *GATA3* expression and contains two non-coding RNA genes that show a similar expression pattern to murine *Dreg1*.

## Discussion

Here, we show that germline excision of the *Dreg1* locus resulted in a specific reduction in ILC2 cells owing to a developmental bottleneck during ILC development in the bone marrow. These results are in line with the fact that very high levels of Gata3 are required for ILC2 development (***Hoyler et al., 2012***; ***Klein Wolterink et al., 2013***). Given its proximity to the Tce1 (+280 kb *Gata3*) enhancer, a surprising finding was that the deletion of *Dreg1* did not affect T or NK cell lineages. While it is tempting to speculate that the *Dreg1* locus represents an ILC2-specific enhancer element, we favour the idea that *Dreg1* likely contributes to dosage tuning of Gata3 during ILC2 commitment and that ILC2s because of their higher Gata3 threshold are more sensitive to modest reductions, whereas T and NK cells are buffered by redundant modules.

Our analysis of publicly available data revealed that the Dreg1 locus was accessible in common lymphoid progenitors and subsequently gained H3K27ac during ILC differentiation coinciding with increased Dreg1 expression. This region was enriched for Tcf1 motifs which was supported by analysis of data from ***Harly et al., 2019***, which showed indeed that Tcf1 bound to these regions in ILCP. Subsequent analysis revealed that H3K27ac of the Dreg1 locus and the expression of Dreg1 was reduced in Tcf1-/- EILP. Taken together, this suggests that Tcf1, which is known to be critical for Gata3 expression

during ILC development is upstream of, and may act in concert with Dreg1 to promote high levels of Gata3 expression in ILC2 development.

While the analysis of publicly available data was invaluable in providing insights into the regulation of Dreg1, further experiments will dig deeper into the mechanisms utilised by Dreg1 to control Gata3 expression. Indeed, how *Dreg1* is involved in regulating *Gata3* levels mechanistically is yet to be determined. *Dreg1* may act as an enhancer RNA that cooperates with chromatin-modifying proteins (*Statello et al., 2021*) to reinforce local H3K27ac and enhancer–promoter communication. Future experiments that selectively suppress *Dreg1* transcription for example via antisense oligonucleotides or CRISPRi at the Dreg1 promoter will enable the discrimination of RNA-dependent from DNA element-dependent effects.

In humans, we identify two non-coding transcripts within the syntenic GATA3 distal enhancer that are highly expressed in ILCP, ILC2, and TH2 cells. Perturbations within this conserved block in an independent tiling-deletion screen performed by *Chen et al., 2023* resulted in reduced GATA3 expression in T cells. While these experiments were performed in T cells, we expect that this enhancer region will likely also play an important role in the regulation of ILC2 development. Given that ILC2 have been implicated in the development of allergic reactions (*Jarick et al., 2022*), the non-coding transcripts we identified may represent targets for the modulation of GATA3 levels and ILC2 function in disease.

## Methods

### Mice

The *Dreg1* knockout mouse line was generated at the Melbourne Advanced Genome Editing Centre (MAGEC) laboratory using CRISPR-Cas9 through microinjection into the pronucleus of one-cell stage C57BL/6 embryos with two sgRNAs (CTACTTGCTGACAAGTCGTC and TCTAGTAAGTCCAGTTGCTT ) to target the 3594 bp genomic region around *Dreg1*. The F0 offspring were validated for *Dreg1* deletion with forward primer GACCAGATATGGAGACGTGCA and reverse primer TCTTTGCCATCTTCTG TGTGC and were backcrossed with wild-type C57BL6/J mice for two or more generations.

Cells and tissues were obtained from littermate control mice aged at least 6 weeks unless otherwise specified. Due to sex differences in ILC populations (*Kadel et al., 2018*), only female mice were used. All mice were maintained at the WEHI Animal Facility under specific pathogen-free conditions. All experiments were approved by The Walter and Eliza Hall Institute Animal Ethics Committee (#2018.002, #2022.002 and #2022.034). Animals were assigned to experimental groups using simple randomisation. Data were statistically analysed by Student's *t*-test using GraphPad Prism, with $p < 0.05$ considered significant.

### Bone marrow chimeras

Mixed bone marrow chimeras were generated by harvesting bone marrow from either wild type (CD45.1[+]) mice or Dreg1-deficient (CD45.2[+]) mice, mixing equally and injecting 5 million cells intravenously into lethally irradiated (2 × 550 Gy) F1 (CD45.1[+]/CD45.2[+]) mice. Mice were left to reconstitute for over 8 weeks.

### Cell isolation

#### Thymus and spleen

The spleen and thymus were made into single-cell suspension by mashing through a 70 µm mesh. Splenic cells were treated with red cell lysis buffer.

#### Bone marrow

Both pairs of hips, femurs, and tibia bones were grounded in a mortar and pestle to collect the bone marrow. The cell suspension was filtered through a 70 µm cell strainer and negatively enriched for ILC cells with the mouse lineage cell depletion kit (Miltenyi).

#### Extraction of lymphocytes from non-lymphoid tissues

Lymphocytes from the VAT and lung were finely chopped in 3–5 mL digestion buffer (RPMI and 2% FCS with 2 mg/mL Collagenase IV). The suspension was digested for 40 min at 37°C and shaken at

180 rpm. Digested VAT samples were topped up to 30 mL with FACS buffer (1× PBS with 2% FCS) and spun at 800 × $g$ for 15 min at 4°C. Lungs were mashed through a 70 mM filter and topped to 30 mL FACS buffer and pelleted at 1700 rpm for 5 min. Both VAT and lung pellets were treated with RBC lysis buffer.

## Isolation of lymphocytes from small intestinal lamina propria

Mesentery and Peyer's patches were removed longitudinally, and faeces removed. Intestine was chopped into 0.5 cm pieces into 1× PBS and vortexed. Intestinal pieces were placed into 20 mL of dissociation buffer (1× HBSS without $Ca^{2+}$ and $Mg^{2+}$ with 2% FCS, 10 mM HEPES and 5 mM EDTA) and digested for 40 min at 37°C at 180 rpm. Digested pieces were washed with FACS buffer and then placed in 5–8 mL of digestion buffer (1× HBSS with indicator containing 2% FCS, 10 mM HEPES, 2 mg/mL Collagenase IV, and 2 mg/mL DNAse I) and incubated for 30–40 min at 37°C at 180 rpm. Digested tissue was mashed through a 70 mM cell strainer and pelleted. A 40/80% percoll gradient was performed to enrich for lymphocytes.

## Flow cytometry and antibodies

Fluorophore conjugated antibodies against mouse antigens were used for flow cytometry. Antibodies, clone names, and manufacturers: **_Biolegend_**: CD4 (GK1.5) APCFire750, CD8a (53–6.7) BV570, NKp46 (29A1.4) PEcy7, ICOS (7E.17G9) BV421, Ki67 (11F6) BV650, CD62L (MEL-14) APC and T-bet (4B10) APC. **_BD_**: CD19 (1D3) BV786, TCRb (H57-597) BV750, TCRgd (GL3) BV711, B220 (RA3-6B2) BV786, GATA3 (L50-823) BUV395, RORgt (Q31-378) BV421, RORgt (Q31-378) PerCPcy5.5, KLRG1 (2F1) BUV615, CD49a (Ha31/8), CD25 (PC61) BV480, NK1.1 (PK136) BUV805, NK1.1 (PK136) BB700, CD44 (IM7) BUV496. **Invitrogen/Bioscience**: CD11c (N418) FITC, FOXP3 (FJK-16s) PE, IL-33R (ST2) (RMST2-2) PerCP-eFLuor710, Streptavidin BUV563. **_WEHI_**: CD45.2 (104) AF700. Live dead staining was performed using Fixable viability stain 440UV diluted in 1xPB at 1:1000 dilution. Surface staining of antibodies was for 30 min on ice at 1/200, 1/300 or 1/400 in FACS buffer. Intracellular staining of transcription factors was overnight at 4°C using the Foxp3/Transcription factor staining buffer set from eBioscience overnight. Cell numbers for specific cell populations were calculated using the volume of the sample that was analysed, the cell counts per sample, and the percentage of the specific cell population.

## RNA-Seq analysis

Publicly available RNA-Seq data were downloaded from SRA and re-analysed. Briefly fastq files were analysed with FASTQC (0.12.1) and adapters were removed using trimmomatic (**_Bolger et al., 2014_**) (0.36). The reads were aligned to GRCm39 for mouse samples and GRCh38 for human samples using hisat2 (**_Kim et al., 2019_**) (2.0.5). Gene counts were quantified using Rsubread (**_Liao et al., 2019_**) featureCounts (1.6.3) with annotation from gencode (**_Mudge et al., 2025_**) (vM33 for mouse and v42 for human), which had the annotation for the _Dreg1_ gene or human homologs of the _Dreg1_ gene added, respectively. Samples were processed using edgeR (**_Robinson et al., 2010_**) (3.42.4). Lowly expressed genes were removed, and samples were normalised according to library size. Gene expression was plotted as logCPM using ggplot2 (3.4.4).

## ATAC-Seq analysis

Publicly available ATAC-Seq data were downloaded from SRA and re-analysed. Processing was based on the ENCODE ATAC-Seq pipeline. Briefly fastq files were analysed with FASTQC and adapters were removed using trimmomatic (0.36). The reads were aligned to the GRCm39 genome for mouse samples and GRCh38 genome for human samples using bowtie2 (2.4.4). Duplicate reads were marked using Picard tools (2.26.11) and reads were filtered using samtools (**_Danecek et al., 2021_**) (1.9) with -F 1804 -f 2. Fold-change coverage tracks were created using macs2 (2.2.7.1) callpeak with a smooth window of 150 and a shift size of –75 and bdgcmp. The bedgraph was converted to a bigwig using ucsc tools (331) bedGraphToBigWig (**_Kent et al., 2010_**) and visualised using IGV (**_Thorvaldsdóttir et al., 2013_**) (2.17.4) with group scaling among samples from the same experiment.

## Motif scanning

The regions of interest as identified through peak calling of ATAC-Seq data were scanned for motifs using FIMO (*Grant et al., 2011*) from the MEME suite (5.0.5) using the Hocomoco (*Kulakovskiy et al., 2018*) (H12) core motif database. The resulting list was filtered to remove factors that were not expressed in ILC2P cells as determined through bulk RNA-Seq (GSE77695) with a cutoff of ≥ 1 LogCPM.

## ChIC and CUT&Run analysis

Publicly available ChIP-Seq or CUT&Run data were downloaded from SRA and re-analysed. Briefly fastq files were analysed with FASTQC and adapters were removed using trimmomatic (0.36). The reads were aligned to the GRCm39 genome for mouse samples using bowtie2 (2.4.4) with the options -q –5 0–3 0. Duplicate reads were marked using Picard tools (2.26.11) and reads were filtered using samtools (1.9) with -F 1804 -f 2. Fold-change coverage tracks were created using macs2 (2.2.7.1) call-peak with a smooth window of 50 and a shift size of –25 for transcription factors and 150 and –75 for histone marks along with macs2 bdgcmp. The bedgraph was converted to a bigwig using ucsc tools (331) bedGraphToBigWig and visualised using IGV (2.17.4) with group scaling among samples from the same experiment.

## CRISPR guides

The guide counts were downloaded from GSE190860 and were used to create a bed file to identify the target regions. The bed file was visualised in IGV (2.17.4).

## Publicly available datasets used in this study

| Dataset type | Accession number |
|---|---|
| Bulk RNA-Seq (Mouse-ILC) (*Shih et al., 2016*) | GSE77695 |
| Bulk RNA-Seq (Mouse-ILC w TCF1KO) (*Harly et al., 2019*) | GSE113767 |
| Bulk RNA-Seq (Human) (*Ercolano et al., 2020*) | PRJEB35186 |
| Bulk ATAC-Seq (ILCP) (*Kasal et al., 2021*) | GSE169542 |
| Bulk ATAC-Seq (HumanMouseILC2) (*Laurie et al., 2024*) | GSE231999 |
| Bulk ChIP-Seq (Histone) (*Gury-BenAri et al., 2016*) | GSE85156 |
| Bulk CUT&Run (EILP TCF1) (*Harly et al., 2019*) | GSE128483 |
| Bulk ChIC-Seq (Histone Tcf1 KO) (*Ren et al., 2022*) | GSE142468 |
| CRISPR Screen (*Chen et al., 2023*) | GSE190860 |

## Acknowledgements

We thank the members of the Allan and Vasanthakumar labs for their input into this project. The authors also gratefully acknowledge the WEHI Flow Cytometry and Bioservices Facilities for their support and assistance in this work. We also thank the Melbourne Advanced Genome Editing Centre (MAGEC) laboratory for generation of the Dreg1[-/-] 349 mice. Funding This work was supported by the NHMRC, Australia Ideas Grant 2001131 and a Stafford Fox Medical Foundation Research Grant to RA and an NHMRC Investigator Grant 2009336 to AV. This work was made possible through Victorian State Government Operational Support Program and the Australian Government NHMRC IRIISS.

# Additional information

## Funding

| Funder | Grant reference number | Author |
|---|---|---|
| National Health and Medical Research Council | 2001131 | Rhys Allan |
| Stafford Fox Medical Research Foundation | | Rhys Allan |
| National Health and Medical Research Council | 2009336 | Ajithkumar Vasanthakumar |

The funders had no role in study design, data collection and interpretation, or the decision to submit the work for publication.

## Author contributions

Sara Quon, Conceptualization, Resources, Data curation, Formal analysis, Supervision, Investigation, Visualization, Methodology, Writing – original draft, Project administration, Writing – review and editing; Adelynn Tang, Conceptualization, Data curation, Formal analysis, Investigation, Methodology, Writing – review and editing; Nadia Iannarella, Data curation, Investigation; Kael Schoffer, Data curation, Formal analysis, Investigation; Wing Fuk Chan, Formal analysis, Methodology; Timothy M Johanson, Formal analysis, Investigation, Writing – review and editing; Ajithkumar Vasanthakumar, Conceptualization, Resources, Data curation, Formal analysis, Supervision, Funding acquisition, Validation, Investigation, Methodology, Writing – original draft, Project administration, Writing – review and editing; Rhys Allan, Conceptualization, Resources, Data curation, Formal analysis, Supervision, Funding acquisition, Investigation, Methodology, Writing – original draft, Project administration, Writing – review and editing

## Author ORCIDs

Sara Quon ⓘD https://orcid.org/0000-0003-3860-3491
Rhys Allan ⓘD https://orcid.org/0000-0003-0906-2980

## Ethics

All experiments were approved by The Walter and Eliza Hall Institute Animal Ethics Committee (#2018.002, #2022.002 and #2022.034).

Reviewer #1 (Public review): https://doi.org/10.7554/eLife.109408.3.sa1
Reviewer #2 (Public review): https://doi.org/10.7554/eLife.109408.3.sa2
Author response https://doi.org/10.7554/eLife.109408.3.sa3

# Additional files

## Supplementary files

MDAR checklist

Source data 1. Source data associated with flow cytometry analyses in *Figures 1 and 2*.

## Data availability

No new datasets were produced in this study. Source data for Figures 1 and 2 has been provided in *Source data 1*.

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
