## [Editor Report · eLife Assessment]

This is a **valuable** study that investigates the role of the long non-coding RNA Dreg1 for the development, differentiation, or maintenance of group 2 ILC (ILC2). The authors generate Dreg1-/- mice and show **solid** evidence for a reduction of group 2 innate lymphoid cells (ILC2). However, the strength of evidence supporting and analysing the impact of Dreg1 on Gata3 expression, a transcription factor required for ILC2 cell fate decisions, remains incomplete. This study will be of interest to immunologists.

---

## [Referee Report · Reviewer #1 (Public review)]

Summary:

This study examines the role of the long non-coding RNA Dreg1 in regulating Gata3 expression and ILC2 development. Using Dreg1 deficient mice, the authors show a selective loss of ILC2s but not T or NK cells, suggesting a lineage-specific requirement for Dreg1. By integrating public chromatin and TF-binding datasets, they propose a Tcf1-Dreg1-Gata3 regulatory axis. The topic is relevant for understanding epigenetic regulation of ILC differentiation.

Strengths:

(1) Clear in vivo evidence for a lineage-specific role of Dreg1.

(2) Comprehensive integration of genomic datasets.

(3) Cross-species comparison linking mouse and human regulatory regions.

Weaknesses:

(1) Mechanistic conclusions remain correlative, relying on public data.

(2) Lack of direct chromatin or transcriptional validation of Tcf1-mediated regulation.

(3) Human enhancer function is not experimentally confirmed.

(4) Insufficient methodological detail and limited mechanistic discussion.

Comments on revisions:

The authors have provided clear evidence that Dreg1 is necessary for ILC2 development, but their refusal to perform any mechanistic experiment remains a significant weakness. While their appeal to the 3Rs and the use of public datasets is noted, re-analyzing external data from heterogeneous sources cannot substitute for direct, internal validation of the Tcf1-Dreg1-Gata3 axis in their specific knockout model. This is particularly problematic because ILC2 progenitors, though rare, can be isolated from bone marrow, especially since assays like CUT&Tag and others are specifically designed for low cell numbers. By relying on public T-cell CRISPR screens to justify human ILC2 functions, the authors are substituting cross-cell-type correlation for definitive functional proof. Consequently, the manuscript currently describes a discovery of necessity without providing a verified molecular mechanism, which should be more explicitly reflected in the title and conclusions.

---

## [Referee Report · Reviewer #2 (Public review)]

The authors investigate the role of the long non-coding RNA Dreg1 for the development, differentiation or maintenance of group 2 ILC (ILC2). Dreg1 is encoded close to the Gata3 locus, a transcription factor implicated in the differentiation of T cells and ILC, and in particular of type 2 immune cells (i.e., Th2 cells and ILC2). The center of the paper is the generation of a Dreg1-deficient mouse. The role of Dreg1 in ILC2 was documented by mixed bone marrow experiments. While Dreg1-/- mice did not show any profound ab T or gd T cell, ILC1, ILC3 and NK cell phenotypes, ILC2 frequencies were reduced in various organs tested (small intestine, lung, visceral adipose tissue). In the bone marrow, immature ILC2 or ILC2 progenitors were reduced whereas a common ILC progenitor was overrepresented suggesting a differentiation block. Using ATAC-seq, the authors find the promoter of Dreg1 is open in early lymphoid progenitors and the acquisition of chromatin accessibility downstream correlates with increased Dreg1 expression in ILC2 progenitors. Examining publicly available Tcf1 CUT&Run data, they find that Tcf1 was specifically bound to the accessible sites of the Dreg1 locus in early innate lymphoid progenitors. Finally, the syntenic region in the human genome contains two non-coding RNA genes with an expression pattern resembling mouse Dreg1.

The topic of the manuscript is interesting. The article is focused on the first description of the Dreg1 knockout mouse and the specific effect of Dreg1 deficiency on ILC2 development.

(1) The data of how Dreg1 contributes to the differentiation and or maintenance of ILC2 is not addressed at a very definitive level. Does Dreg1 affect Gata3 expression, mRNA stability or turnover in ILC2? Previous work of the authors indicated that knock-down of Dreg1 does not affect Gata3 expression (PMID: 32970351). The current data (Figure 2H) showed small differences in Gata3 expression in CHILP which were, however, not statistically significant. No differences were found in ILCP and ILC2P.

(2) How Dreg1 exactly affects ILC2 differentiation remains unclear.

---

## [Author Response]

The following is the authors’ response to the original reviews

**Public Reviews:**

**Reviewer #1 (Public review):**
Summary:This study examines the role of the long non-coding RNA Dreg1 in regulating Gata3 expression and ILC2 development. Using Dreg1-deficient mice, the authors show a selective loss of ILC2s but not T or NK cells, suggesting a lineage-specific requirement for Dreg1. By integrating public chromatin and TF-binding datasets, they propose a Tcf1-Dreg1-Gata3 regulatory axis. The topic is relevant for understanding epigenetic regulation of ILC differentiation.Strengths:(1) Clear in vivo evidence for a lineage-specific role of Dreg1.(2) Comprehensive integration of genomic datasets.(3) Cross-species comparison linking mouse and human regulatory regions.Weaknesses:(1) Mechanistic conclusions remain correlative, relying on public data.

We agree that the mechanistic conclusions are of our study are indeed correlative and we mention this in the discussion. The primary work of the study is the discovery of *Dreg1*'s necessity for ILC2 development via the new knockout mouse model. Re-analysing good quality publicly available data on rare cell populations is an appropriate approach and in line with DORA guidelines for ethical research.

(2) Lack of direct chromatin or transcriptional validation of Tcf1-mediated regulation.

The most appropriate way to examine direct Tcf1 target genes in primary cells is to examine the association of Tcf1 binding with the changes that occur in Tcf1-bound genes after *Tcf7* knockout. By analysing publicly available data on ILC progenitors we indeed did this. We revealed that Tcf1 bound to *Dreg1* and that *Dreg1* was not expressed when Tcf1 was knocked out in ILC progenitors. In addition we examined H3K27ac at the *Dreg1* locus in the same ILC progenitors to demonstrate that Tcf1 appears to be important for decorating the *Dreg1* gene with this histone modification. We believe that this analysis is sufficient to conclude that Tcf1 is required for the expression of *Dreg1* in ILC progenitors.

(3) Human enhancer function is not experimentally confirmed.

We agree that the potential human enhancer of GATA3 we identified has not been confirmed in human ILC. However, a previous study showed clear evidence that this region has GATA3 enhancer activity in human T cells. Therefore, while not specific to ILC2s the region where the DREG1 homologues lie does indeed harbour enhancer activity.

(4) Insufficient methodological detail and limited mechanistic discussion.

We have now made the changes suggested by the reviewer to both the methods/figure legends and also the discussion.

**Reviewer #1 (Recommendations for the authors):**
The authors generated Dreg1-deficient mice and demonstrated that loss of this locus selectively reduces ILC2s but not T or NK cells, indicating a lineage-specific requirement for Dreg1 in ILC development. By analyzing publicly available chromatin accessibility and transcription factor-binding datasets, they link Dreg1 expression to Tcf1-dependent chromatin activation and extend their findings to human data by identifying a syntenic GATA3 enhancer that produces homologous Dreg lncRNAs in ILC2s. While the study addresses an interesting question, most of the mechanistic interpretations rely heavily on publicly available datasets rather than the authors' own functional evidence. To establish causality and reinforce the overall conclusions, I provide below some comments and suggestions for additional experiments and clarifications that would considerably strengthen the manuscript.(1) In Figure 3, the authors use public datasets to argue that Tcf1 regulates Dreg1 expression by modulating chromatin accessibility and H3K27ac at its locus. However, since these data are derived from heterogeneous external sources, the conclusions remain associative. To better support causality, the authors should generate matched datasets from their own sorted progenitor populations and perform CUT&Tag for Tcf1 and H3K27ac in wild-type and Tcf7 knockout progenitors to directly test whether Tcf1 binding establishes an active chromatin state at Dreg1. Also, complementing this with nascent RNA or pre-mRNA quantification would link chromatin activation to transcriptional output. These experiments are technically feasible in progenitors and would substantially strengthen the claim that Tcf1 directly drives Dreg1 activation during ILC development.

We believe that utilising publicly available data sufficiently answers this question while also adhering to ethical considerations. The ILC populations used to produce the publicly available data were akin to those we examined in our analyses, and the data was of sufficient quality. Moreover, they enable us to access data from Tcf1-deficient mice. Redoing large-scale chromatin profiling on rare cell types would require hundreds of mice to achieve sufficient cell numbers. Repeating this solely for “originality” contradicts the 3Rs principles (replacement, reduction, refinement) if high quality public data already exists and we feel will require years of redundant work. In addition, we believe the fact that the data derive from heterogenous external sources, yet align well, only strengthen our conclusions. We have now added mention to our use of publicly available data in the discussion.

(2) In Figure 4, the authors provide correlative evidence from public datasets suggesting that the human region syntenic to the murine Dreg1 locus acts as a distal enhancer of GATA3 and gives rise to two ILC2-specific lncRNAs. To substantiate this claim, the authors should perform CUT&Tag for H3K27ac in human ILC2s to confirm enhancer activation and use 3C or HiChIP to demonstrate physical interaction with the GATA3 promoter. These experiments should be doable by fusing pooled ILC2 samples and would provide more direct evidence that this region actively regulates GATA3 expression.

Assessing the activity of a distal enhancer region on its target gene in primary human cells is extremely difficult, due to a number of technical and biological complications such as enhancer redundancy. This is why we chose to reanalyse an extensive enhancer deletion screen performed in human T cells by Chen et al., AJHG 2023. This analysis clearly showed deletion of the region we identified as harbouring *Dreg1* homologues affected GATA3 expression, thus confirming its enhancer activity. While we agree with the reviewer that specific profiling of human ILC populations for H3K27ac and 3D genome architecture would provide further correlative evidence this will be a time-consuming and costly endevour with human material and ultimately the definitive proof in ILCs would require specific deletion of this region in ILC2s. We have mentioned this caveat in the discussion.

(3) Several figure legends lack essential methodological details. Figure 1 should specify how NK and ILC populations were gated, including intermediate steps and markers used. The same applies to Supplementary Figure 1, and particularly to Supplementary Figure 2, where gating strategies for progenitors are shown but not explained. Figure 2 should also indicate that these analyses were performed in bone marrow. Clearer legends are crucial for interpreting and reproducing the data.

We have made the suggested changes.

(4) It is also unclear throughout the manuscript whether the authors performed any ATACseq experiments themselves or relied entirely on public datasets. This information should be stated explicitly in the main text and figure legends, not only in the Methods section. Similarly, the source of the ChIPseq or CUT&Run datasets should be clearly indicated alongside the relevant figures.

We apologise for not making this clearer and have now clearly articulated if the data was public in the text.

(5) As the authors themselves suggest, performing experiments that selectively suppress Dreg1 transcription using antisense oligonucleotides or CRISPR interference at the Dreg1 promoter would provide more valuable mechanistic insights. Conducting these experiments in their own system would allow them to determine whether Dreg1 functions through its RNA product or as a DNA enhancer element, thereby strengthening the causal link between Dreg1 activity and Gata3 regulation.

We agree with the reviewer, however, this, in our opinion is beyond the scope of this manuscript. The strength of this manuscript lies in the findings from the novel *Dreg1* knockout mouse strain. Future studies will focus on understanding how *Dreg1* influences Gata3 expression.

(6) The discussion would benefit from a clearer and more integrated explanation of how Dreg1 fits into the transcriptional network that controls ILC2 differentiation. The authors could elaborate on whether Dreg1 fine-tunes Gata3 expression or functions as part of a regulatory loop with Tcf1, and better explain how this mechanism might be conserved in humans. In addition, the authors should explicitly acknowledge the limitations of relying on publicly available datasets and emphasize the need for direct experimental validation to support their mechanistic interpretation.

We have now made these suggested inclusions.

**Reviewer #2 (Public review):**
The authors investigate the role of the long non-coding RNA Dreg1 for the development, differentiation, or maintenance of group 2 ILC (ILC2). Dreg1 is encoded close to the Gata3 locus, a transcription factor implicated in the differentiation of T cells and ILC, and in particular of type 2 immune cells (i.e., Th2 cells and ILC2). The center of the paper is the generation of a Dreg1-deficient mouse. While Dreg1-/- mice did not show any profound ab T or gd T cell, ILC1, ILC3, and NK cell phenotypes, ILC2 frequencies were reduced in various organs tested (small intestine, lung, visceral adipose tissue). In the bone marrow, immature ILC2 or ILC2 progenitors were reduced, whereas a common ILC progenitor was overrepresented, suggesting a differentiation block. Using ATAC-seq, the authors find that the promoter of Dreg1 is open in early lymphoid progenitors, and the acquisition of chromatin accessibility downstream correlates with increased Dreg1 expression in ILC2 progenitors. Examining publicly available Tcf1 CUT&Run data, they find that Tcf1 was specifically bound to the accessible sites of the Dreg1 locus in early innate lymphoid progenitors. Finally, the syntenic region in the human genome contains two non-coding RNA genes with an expression pattern resembling mouse Dreg1.The topic of the manuscript is interesting. However, there are various limitations that are summarized below.(1) The authors generated a new mouse model. The strategy should be better described, including the genetic background of the initially microinjected material. How many generations was the targeted offspring backcrossed to C57BL/6J?

The mice were backcrossed for at least 2 generations to C57BL/6. This information is now included in the methods section.

(2) The data is obtained from mice in which the Dreg1 gene is deleted in all cells. A cell-intrinsic role of Dreg1 in ILC2 has not been demonstrated. It should be shown that Dreg1 is required in ILC2 and their progenitors.

We now provide new mixed bone marrow irradiation chimera data that shows that the effect is intrinsic to *Dreg1*-deficient ILC2 cells (Figure 1F and Supplementary Figure 1E-G).

(3) The data on how Dreg1 contributes to the differentiation and or maintenance of ILC2 is not addressed at a very definitive level. Does Dreg1 affect Gata3 expression, mRNA stability, or turnover in ILC2? Previous work of the authors indicated that knockdown of Dreg1 does not affect Gata3 expression (PMID: 32970351).

We have indeed shown that *Dreg1*-deficient ILC2P have reduced levels of Gata3 (Figure 2H) however we have not determined the exact mechanisms by which *Dreg1* controls ILC2 development.

(4) How Dreg1 exactly affects ILC2 differentiation remains unclear.

We agree with the reviewer, however, this article is focused on the first description of the *Dreg1* knockout mice and the surprisingly specific effect on ILC2 development.

**Reviewer #2 (Recommendations for the authors):**
(1) Relating to point 2 of public review:It should be shown that Dreg1 is required in ILC2 and their progenitors. Mixed bone marrow chimeras would be an adequate strategy.

We have now done this and clearly showed that the effect is intrinsic to *Dreg1*-deficient ILC2s.

(2) Relating to point 3 of public review:Minimally, Gata3 expression should be analyzed in ILC2, ILC2P, and the ILC progenitors by qRT-PCR and antibody stain.

We have indeed shown reduced Gata3 levels by antibody stain in Figure 2H.

(3) Relating to point 4 of public review:The manuscript would benefit from additional data studying ILC2 differentiation in (competitive) adoptive transfer experiments or using in vitro differentiation assays.

We have performed the mixed bone marrow chimera experiments which are testing the competitiveness of *Dreg1*-deficient bone barrow with control wildtype. In this case the WT ILC2s outcompeted the *Dreg1*-deficient ILC2s for the same niche.